# Patient Input in Regional Healthcare Planning—A Meaningful Contribution

**DOI:** 10.3390/ijerph16193754

**Published:** 2019-10-05

**Authors:** Heidrun Sturm, Miriam Colombo, Teresa Hebeiss, Stefanie Joos, Roland Koch

**Affiliations:** University Hospital Tübingen, Institute for General Practice and Interprofessional Care, Osianderstraße 5, 72076 Tübingen, Germany; heidrun.sturm@med.uni-tuebingen.de (H.S.); miriam.colombo@med.uni-tuebingen.de (M.C.); teresa.hermann@gmx.net (T.H.); stefanie.joos@med.uni-tuebingen.de (S.J.)

**Keywords:** healthcare planning, regional disparities, primary care, patient satisfaction, citizens’ perception, community, health services research, questionnaires, patient needs

## Abstract

*Background:* There are well-known methodological and analytical challenges in planning regional healthcare services (HCS). Increasingly, the need for data-derived planning, including user-perspectives, is discussed. This study aims to better understand the possible contribution of citizen experience in the assessment of regional HCS needs in two regions of Germany. *Methods:* We conducted a written survey in two regions of differing size—a community (3653 inhabitants) and a county (165,211 inhabitants). Multinomial logistic regression was used to assess the impact of sociodemographic and regional factors on the assessment of HCS provided by general practitioners (GPs) and specialists. *Results:* Except for age and financial resources available for one’s own health, populations did not differ significantly between the regions. However, citizens’ perception of HCS (measured by satisfaction with 1 = very good to 5 = very poor) differed clearly between different services (e.g., specialists: 3.8–4.3 and pharmacies: 1.7–2.5) as well as between regions (GPs: 1.7–3.1; therapists: 2.9–4). In the multivariate model, region (next to income and age) was a consistent predictor of the perception of GP- and specialist-provided care. *Discussion:* Citizens’ perceptions of HCS correspond to regional provider density (the greater the density, the better the perception) and add insights into citizens’ needs. Therefore, they can provide valuable information on regional HCS strengths and weaknesses and are a valid resource to support decision makers in shaping regional care structures.

## 1. Introduction

Germany, like many other countries, is facing a pronounced demographic change that also affects healthcare service (HCS) demand [1,2]. Elderly people have a greater need of care [3] and, due to multimorbidity, often the need is for more coordinated support [2]. At the same time, rural and urban areas in Germany (as in most other countries) show disparities in population and infrastructure including in healthcare services [4,5]. For example, a gradient in care provider density has been revealed between the northeast region of Mecklenburg-Vorpommern to the southwest region of Baden-Württemberg [6,7].

Both care provider age and consumer age tend to be higher in rural areas, while population and provider density tend to be lower. Aside from varying geographic and demographic characteristics, there are differences with respect to access to social services, infrastructure, and daily life necessities, such as pharmacies, shopping opportunities or access to financial services [8,9].

These prerequisites affect the demands on HCS, especially in rural areas. Thus, HCS planning urgently needs to adapt to the emerging situation.

According to the recommendations of the “Expert Council for the Evaluation of the Development in the Health Service”, local demographic and socioeconomic factors should be considered for regional healthcare planning [7]. German legislation had acknowledged this development with the “Law for the Improvement of Care Structures in the Statutory Health Insurance” (GKV-VStG); the first of a series of legislation that was passed in 2011 [10]. This law introduced more flexibility for care organization as well as regional planning and, in theory, allowed for a greater degree of communal participation in the regional planning process. The law also included reform of the care planning process for outpatient care (Bedarfsplanungs-Richtlinie [11]), emphasizing the importance of regional characteristics. However, strict common federal rules continued to apply, based mainly on a supplier–population relationship. Responsibility for the implementation of the legislation fell to the regional Association of Statutory Health Insurance Physicians (a body representing physicians in private practices). Therefore, regional particularities were often not sufficiently considered. This policy has recently been updated introducing a new complex measure that takes morbidity and mortality into account [12]. However, the new regulation neglects provider age. Available data on regional variation of providers and therapeutic procedures [13] tend to fall short in sufficiently capturing the preferences of the regional population.

In the past few years, many German communities established local health conferences, which also focus on regional care planning. An analysis of the regional care structure is often conducted at the onset of such conferences. In addition to available statistical data on population, providers, and therapeutic procedures, perceptions of regional HC professionals and/or of the citizens are frequently included [14,15].

Patient dissatisfaction appears to be associated with poor clinical outcomes and increased costs [16]. A recent study found differences in patient satisfaction when comparing hospital and post-hospital discharge surveys. Additionally, patients with selected characteristics (e.g., lower educational status or older age) reported higher patient satisfaction, suggesting that socioeconomic characteristics need to be considered [17]. International studies have shown that consumer participation in regional health planning can have a positive impact on the local health situation, especially in terms of needs assessment and possible solutions [18,19].

In Ontario, Canada, for instance, regular consumer surveys serve as a quality management tool and as guidance for service development [20]. Thus, local governments or communal mayors receive significant input concerning care planning based on citizen feedback and local needs. Furthermore, organizations use consumer satisfaction surveys for planning, especially in primary care [21], and established instruments are available [22].

Patients’ or citizens’ perceptions on regional HC structures have been used in different settings [23,24,25]. Previous studies examined the value of communal input for regional HC planning in Germany which until now mostly relied on the input of local governments [26,27,28,29,30]. To our knowledge, usability and credibility of citizens’ views, such as satisfaction with regional HC services in the context of HC planning is not well examined [31,32].

The aim of this study was to examine citizens’ perception on regional healthcare structures. By comparing different regions, we aim to examine whether regional specifics are reflected in the local citizens’ perception. We hypothesize that citizens’ views on their HC services can provide valuable insights to improve regional HC infrastructure.

## 2. Materials and Methods

### 2.1. Study Population and Data Collection

This cross-sectional study is based on two anonymous surveys, which were conducted in two different German regions located within the federal state of Baden-Württemberg, Germany. The same questionnaire was used for both surveys. Survey A was conducted in a single rural community (Community A) of approximately 3000 inhabitants located *within* County A (population 284,082; median age 45.2 years). Survey B was conducted in an *entire* county, County B (total population of 168,852; median age 45.8 years). Counties A and B both comprise rural areas and industrialized centers. Both surveys were part of regional care structure analyses; in Community A this analysis was requested in the context of the planning of a primary care center, whereas survey B served as a starting point to bring the precarious healthcare structure into the political agenda. Survey A was conducted in May 2016 as a complete survey of Community A and comprised all adult inhabitants (*n* = 2918; total number of inhabitants of Community A: *n* = 3653). Participants were recruited by letter and advertisements in the municipal newspaper with a return rate of 32.8% (*n* = 956). Survey B was conducted in November 2016 using a randomly selected sample of 1000 adult citizens of County B. The return rate was 27.8% (*n* = 278).

The questionnaire was developed based on the established Canadian Index on Wellbeing [20] and on prior interviews with regional stakeholders. It was tested using the “think aloud” method to detect potential unclear wording leading to unintended interpretations [33] and piloted the questionnaire with eight volunteers of different age, sex, and background. The questionnaire comprised four domains: demographics (age, sex, educational status, and monthly income), self-assessed (mental and physical) health status, and health-associated resources (time to take care of one’s health, support by friends and family, financial resources), assessment of regional infrastructure and assessment of healthcare services (HCS provided by general practitioners (GPs), specialists, apothecaries, physiotherapists, psychotherapists, etc.). Health-associated resources, regional infrastructure and HCS were rated using a five-point Likert scale with a lower number corresponding to higher satisfaction (1 = very good, 2 = good, 3 = satisfactory, 4 = poor, 5 = very poor).

Regional characteristics (population density, number of physicians per 1000 inhabitants) were sourced from the Federal Statistical Office of Germany and the regional Association of Statutory Health Insurance Physicians [34,35].

To account for the size of communities, we divided Region B into two groups (communities with less than 7000 inhabitants and communities with 7000 or more inhabitants). See also tables 1 and 2 for reference.

Positive ratings (very good/good) were amalgamated and negative ratings (very poor/poor) likewise, thus forming three categories (very good/good, satisfactory, very poor/poor). Furthermore, the variable, *age*, was divided into the following categories: 18–39, 40–65, and ≥66 years. The variable, *income*, was also divided into three categories (<1500€, 1500≤3000€, and ≥3000€ per month).

### 2.2. Statistical Analysis

Variables were tested for differences between regions using the Kruskal–Wallis test for ordinal variables and the chi-square test for nominal variables, respectively. To examine the influence of population-related predictors, we conducted multinomial logistic regression models for two outcomes: (1) citizens’ perception of HCS provided by GPs and (2) citizens’ perception of HCS provided by specialists, respectively. For both outcomes, possible predictors were identified based on the existing literature [17,20,36,37,38,39]. Variables were included in the multivariate multinomial logistic regression models if they proved to be statistically significant. Variables tested in the univariate model and excluded for GPs were sex, education, employment status, presence of any chronic disease (self-reported), evaluation of traffic infrastructure, evaluation of one’s physical and mental health status (Table 3). Additional exclusions for specialists were evaluation of own financial resources for preventive healthcare and evaluation of support from friends and family (Table 4).

From a total of *n* = 1234 observations, we excluded those with missing or implausible data (e.g., younger than 18 years or living in regions/communities other than A or B) for the variables age, sex, or region (*n* = 131).

Cases with missing outcome variables were excluded. For “evaluation of HCS by GP” *n* = 47, for “evaluation of HCS by specialists” *n* = 127 were excluded. After excluding missing data, the final study population for the outcome “evaluation of HCS by GP” resulted in 1056 participants and for the outcome “evaluation of HCS by specialists” in 976 participants, respectively.

Microsoft Excel Version 14 (© Microsoft 2010, Redmond, Washington, United States of America) and SPSS Version 25 (© IBM 2017, Armonk, New York, United States of America) were used for data processing and analysis.

The study was conducted in accordance with the Declaration of Helsinki. According to the ethics committee of the faculty of medicine of the University of Tübingen, an ethics approval was not needed in order to collect anonymous data according to §3 Federal Data Protection Act/State Data Protection Act Baden-Württemberg (Correspondence 697/2014VF).

## 3. Results

### 3.1. Study Regions and Population

Geographical and population characteristics of Regions A, B1, and B2 are presented in Table 1. To distinguish between different community sizes, Region B was further subdivided into Regions B1 and B2. Region B1 comprised communities in Region B with less than 7000 inhabitants, whereas Region B2 comprised communities with 7000 inhabitants or more. The highest GP density was found in Region A, which was also reflected in the GP supply rate (a measure that is used to restrict the number of new GP practices in one district, if the supply rate exceeds 110%). The specialist density was lower in regions with fewer inhabitants (Region A, Region B1) compared to communities with more inhabitants (Region B2), however, there are communities without any specialist or GP in both Regions B. However, there were less undersupplied specialist groups in Region A (one out of 13 specialist groups) than in Regions B1 and B2 (six out of 13 specialist groups; see Table 1).

Characteristics of the study population are presented in Table 2. Study participants in community A, Region B1, and Region B2 differed significantly from each other in terms of age, and those in community A were significantly older than those in Region B1 and B2. Furthermore, study participants in Community A rated their available financial resources to take care of their own health less favorably than those in Regions B1 and B2 (see Table 2). The study population in the three regions/communities did not differ significantly from each other when it came to other sociodemographic or health-related characteristics.

### 3.2. Differences of HCS Ratings across Regions

Figure 1 shows citizens’ evaluation of different HCS in community A and Regions B1 and B2. Across regions/communities, ratings of psychotherapists tended to be lower than ratings of other healthcare services, such as those of pharmacies, dentists, GPs, as well as homecare services. However, the official psychotherapist supply rate of 129% in Region A and 115% in Regions B1 and B2 indicated no actual lack of psychotherapists (see Table 1). Furthermore, ratings of specialist physicians were low in community A as well as in Regions B1 and B2, with still differing scores between the three tested regions. Additionally, naturopaths, occupational therapists, and other counselling services were also rated less favorably.

Ratings of HCS differed significantly between the regions/communities except for those of psychotherapists and home care services (see Figure 1). Study participants in Community A were markedly more content (mean 1.7) with HCS provided by GPs than those in Region B1 (mean: 3.1) and B2 (mean: 3.1), respectively. Specialist physicians (such as obstetrics and gynecology, ear–nose–throat, dermatologists, orthopedics and anesthesiologists) were slightly better rated in Region B2 (mean 3.8) with bigger communities (≥7000 inhabitants) than in the other two regions/communities (Community A: mean 4.0; Region B1: mean 4.3, (*p* < 0.05)). Ratings of community A were somewhat better than in rural Region B, potentially reflecting the poorer overall specialist supply rate in County B (the supply rate of specialist physicians was below the defined need in six out of 13 specialties in the county of regions B1 and B2, but only in one out of 13 in the county of community A (see Table 1) [35].

### 3.3. Influence of Regional and Socioeconomic Characteristics on Ratings of HCS

To better understand the influence of regional and socioeconomic factors, multivariate multinomial logistic regression analyses were conducted. The outcomes were 1) ratings of HCS by GPs and 2) ratings of HCS by specialist physicians. Results of the multivariate regression analyses are presented in Table 3 and Table 4, respectively.

#### 3.3.1. Ratings of HCS Provided by GPs

For outcome (1) rating of HCS provided by GPs, the variables *attribution to region, age, monthly income, and evaluation of one’s own financial resources* as well as evaluation of *support by friends and family* proved to be statistically significant in univariate regression and were therefore included in the multivariate regression model (see Table 3). *Attribution to region* was significantly associated with ratings of HCS, despite the adjustment for other variables. Compared with study participants from Community A, those in Region B1 and B2 were consistently less likely to give very good/good and satisfactory ratings, respectively. In other words, citizens of Region A consistently rated GPs better than citizens of both Region B1 and Region B2 (*p* < 0.01) independent of age, monthly income, and available financial and personal resources (Table 3). Furthermore, younger study participants (18–39 and 40–65 years) were more likely to rate HCS provided by GPs with “satisfactory” instead of “very good/good” compared to older study participants (≥66 years). Additionally, those with a monthly income below 1500€ gave significantly better ratings for HCS provided by GPs compared to those with an income of 3000€ per month or more.

#### 3.3.2. Ratings of HCS Provided by Specialist Physicians

The variables *attribution to region*, *age*, and *monthly income* were included in the multivariate model for outcome 2) ratings of HCS provided by specialist physicians. There were no statistically significant differences in ratings between Region B2 compared to Community A (see Table 4). However, study participants in Region B1 were significantly more likely to rate HCS provided by specialist physicians with very good/good and satisfactory instead of very poor/poor compared to those in Community A. Furthermore, those with a monthly income below 1500€ or between 1500 and <3000€ also tended to give better ratings to HCS provided by specialist physicians. Age did not seem to significantly affect ratings of specialist physicians independent of region and monthly income.

Citizens’ ratings of specialists are similar in small communities (Region A and B1), while their rating of specialists in the more urban communities of Region B2 is significantly higher than in Region A (Table 4).

## 4. Discussion

The goal of our study was to understand what role citizen surveys can play in the context of regional healthcare services analyses and planning. The surveys yielded an acceptable response rate and ratings showed clear differences between regions and healthcare domains. In multinomial regression, *attribution to region* and *income* had a substantial impact on the evaluation of HCS—both for GP and for specialists. By comparing two geographically different regions in Southern Germany, we could show that citizens’ perceptions of provided care structures are not explained by population characteristics alone but rather seem to distinctly discriminate between services provided. This suggests that citizens’ views provide valuable insights for HC planners.

### 4.1. Study Population and Self-Reported Health

To account for community size, Region B was split into two groups. As expected, provider density was higher in regions with more inhabitants.

The three regions did not differ from each other in terms of sociodemographic characteristics of the respondents except for age and private financial resources available for health, both of which were significant predictors for the rating also in the multivariate model for GPs.

Self-reported morbidity was similar in all regions and is in line with a recent population survey, where 41 percent of respondents from Baden-Württemberg reported a chronic disease [41]. Having a chronic disease did not influence perception of healthcare. Self-reported health is established as an independent predictor of HC consumption in Germany [42] and correlates with symptoms [43]. Therefore, subjective ratings of the patients’ or citizens’ own health have been shown to be reliable.

### 4.2. Specific Rating of Different Healthcare Providers

Citizens in this study had a distinct perception of different providers: Ratings differed clearly between provider groups and showed unique profiles for each region. GPs, pharmacies, and dentists consistently showed a better rating across regions as compared to psychotherapists and naturopaths.

This can be explained by provider density. For example, dentist ratings were consistently better than for other services, but still differed by regional density. This also underlines the assumption that citizens’ perception reflects on the local situation. Moreover, general trends independent of provider density could be detected: Psychotherapists were rated badly despite their formally sufficient density. Waiting times, however, are still too long [44]. Naturopaths and complementary medicine providers are popular in Germany [45] but there are no regulations with respect to planning. To this end, citizens’ experience provides additional information for planners and helps to adjust formal planning mechanisms or regional deficits. In the new planning regulation, the formal threshold of psychotherapists was already lowered due to long waiting times [46].

### 4.3. Factors Influencing Perception of Healthcare Quality

#### 4.3.1. Age: Young Citizens Tend to Give Lower Ratings

Our study showed a weak influence of *age*. There was a tendency toward more criticism by younger people of HCS provided by GPs. This is in line with the known general trend that older people rate HCS better than younger people [17,38].

#### 4.3.2. Income: Lower Income and Better Rating

With the exception of age and income, no sociodemographic factors influenced perception of care in our analysis. The general tendency toward more positive ratings by citizens with lower income might seem surprising, as health status declines with social status [47,48]. Accordingly, HC consumption is higher in this group, which in turn generates more experiences with HCS [49].

Generally, there is good access to GP and SP in Germany due to statutory healthcare insurance and low co-payments [50], which should ease the effect of low income. Furthermore, access to specialists is not restricted by gatekeeping, so everyone has free access to specialist care without the need to consult a GP first. However, data from Germany show that persons with a lower socioeconomic status (SES) are more likely to consult a general practitioner than a specialist compared to persons with a higher SES [49]. A possible explanation is that people with a lower SES are less demanding and, therefore, might also be less critical. This has been described in a more general context [51] and is consistent with the fact that expectations shape satisfaction [52]. Patient satisfaction research has shown that patients rarely can change their situation. Because of that patients adapt cognitively, which leads to a resigned pseudo-satisfaction. Patients’ dependency on HC providers and information asymmetry further contribute to this phenomenon [52]. In a hospital context, persons with a significant disease burden perceive low empowerment *and* paradoxically give better ratings of their care. Our data suggest that citizens’ perception in an outpatient setting follows similar patterns as patient satisfaction in hospitals.

#### 4.3.3. Region Has the Strongest Influence

We found substantial evidence that place of residence and therefore regional experiences with healthcare, played an important role in the evaluation of HCS. This regional influence is not entirely explained by population characteristics.

The regional differences in our study reflect differences in physician density [10,24]: Region A had the highest GP:inhabitant ratio (0.8) suggesting better availability and, therefore, better experiences with care. This corresponded to a significantly better evaluation of GP-based HCS in Region A than in both regions of B. Accordingly, rating and specialist density in Region A was similar to Region B1 (with communities up to 7000 inhabitants), while the rather urban Region B2, with the highest specialist density, showed the best rating of specialist HCS.

On the other hand, the overall specialist supply rate and, therefore, availability in the entire Region B was markedly lower than in the region to which community A belongs. However, HCS ratings did not differ significantly. Pure physician-to-population ratios of defined regions are often criticized for neither representing actual catchment areas of GPs nor taking population needs into account [4]. In order to improve this measure, a new planning factor (gravitation index) has been developed recently for Germany, which adjusts for “regional needs and provided offers” including care provided by neighboring regions and population specifics such as age and morbidity [12]. In the report that provided the baseline for a new planning structure in German outpatient care [46], time needed to reach the nearest provider proved to depend largely on community size. Yet access and waiting times depend on physician density, patient preferences, and sociodemographic factors [12].

In our study, citizens’ ratings of HCS of GPs and specialists reflect regional provider density largely independent of their personal setting. This suggests that regional provider density provides central information for regional HC planning and, at the same time, correlates with user experience to a certain degree. However, relying on quantitative measures alone does not fully capture regional needs.

#### 4.3.4. Citizens Can Provide Important Additional Insights for Planning

Our results support the notion that citizens’ perception of HC provision is a valid indicator in regional care planning. First, different care providers were rated distinctly different, indicating that citizens do not generally judge their entire regional care positively or negatively but perceive clear differences according to their experiences. In addition, perceived care by GPs and specialists seems to correlate with physician density.

Since administrative planning measures can always only capture certain aspects, additional local data can enhance need-based planning. Even with the new German planning regulation, being implemented in fall 2019 and resulting in a significant number of open practice locations, provider age is not captured. However, this is vital when estimating provider availability for the future. Furthermore, valid quantitative data is not readily available for all providers (e.g., physiotherapists, homecare) and those data available refer to large regions comprising several communities, thus not reflecting local needs. The combination of formal HCS planning (such as physician density, see above) and patient experience/satisfaction can lead to a mutual validation in this context, providing a safer ground to start from.

Consumer experience is established in various settings in healthcare, mostly to support individual or patient-centered care [32]. Patient experience is also used in international [53] or institutional settings [54] to detect deficits or to help improve care experiences. Evidence concerning its use for regional care planning is scarce [19,26,55]. Based on our results, it can be concluded that community-based patient input, such as satisfaction with regional healthcare services, enhances the overall quality of regional healthcare service planning and research.

### 4.4. Limitations

Our results show a satisfactory external validity when compared with regional administrative reference data. Nevertheless, our study focus is limited to two defined regions in Germany and, therefore, needs to be validated in different settings and regions. Citizens’ responses showed plausible levels of, for example, chronic disease or employment rate when compared with available data.

The adaptation of an established questionnaire and the piloting phase with regional stakeholders, as well as a conservative approach to statistical analysis contributed to the internal validity of our study.

The major limitation of this study is the comparison of two regions: While Region A is one community, Region B is a larger county that includes many separate communities. To increase comparability and to account for community size, we divided the communities from Region B into two subgroups based on the number of inhabitants. Additionally, sample sizes of the two surveys differed: In community A, all adults were invited to participate (*n* = 2918). In Region B, a representative sample of 1000 inhabitants was surveyed. This led to different sample sizes (*n* = 921 and *n* = 209), which can potentially inflate statistical effects and are susceptible to selection bias.

Despite the limitations, our results contribute to a more differentiated understanding of the contribution of consumer surveys concerning subjective assessment of HCS in regional health planning.

### 4.5. Future Prospects

One of the greatest challenges in local healthcare service research may be the validity of available data. Our data suggest that citizens’ satisfaction with care can support regional planning. Actual regional care planning is based on care supply and will include consumer characteristics. However, these factors seem to only predict a part of actual care consumption—at least in inpatient care [56]. Patient preferences added only little to the explanation of unclear variation in care consumption [26]. However, including consumer satisfaction in co-production of healthcare has been suggested to “[…] improve public services by (1) employing the expertise of service users and their networks; (2) enabling more differentiated services and more choice for service users; (3) increasing responsiveness to dynamic user need and (4) reducing waste and cost” [57]. Therefore, we propose to combine population/patient survey data with secondary data from both official statistical sources and service providers (e.g., healthcare insurances) to support regional HCS planning. By doing so, two different perspectives would be included, one representing attitudes and the other actual behavior, offering a more detailed picture of regional care needs. In addition, qualitative approaches might further specify user-perception and needs [58].

Furthermore, perceived care quality should be extended to physicians and the interaction between physicians, allied health professions, and patients. In addition to strengthening evidence of local care deficits, this could help to understand the influence of physician recommendations in healthcare utilization [26].

## 5. Conclusions

Citizens’ perceptions of HCS correspond with regional characteristics, especially provider density. We conclude that patient surveys targeting the assessment of HCS provide valuable information about regional HCS characteristics and patient needs. It should be considered an important puzzle piece in regional HCS assessment and planning to meet the demands of the population.

## Figures and Tables

**Figure 1 ijerph-16-03754-f001:**
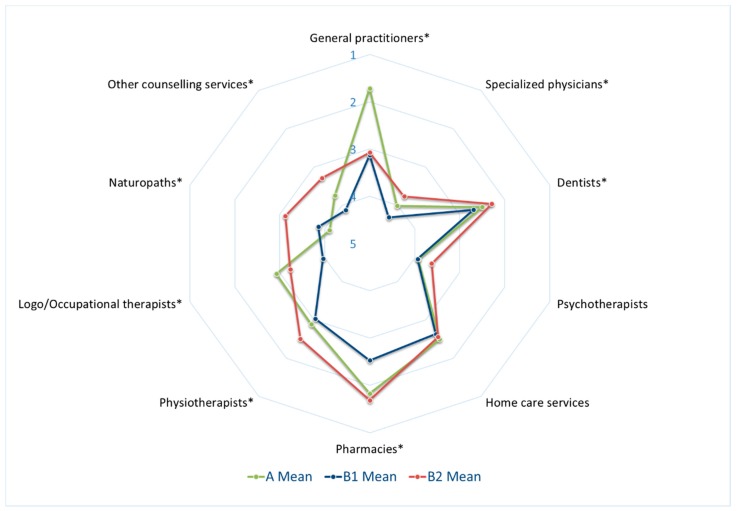
Perception of different healthcare structures by region/community. Average ratings of community A are displayed in green, those of Regions B1 and B2 in blue and red, respectively. Presented are means of scales. Study participants were asked “How do you rate healthcare services offered by …?”) and rated their answers from 1 = very good to 5 = very poor (*Significant statistical difference between groups (*p* < 0.05), assessed using Kruskal–Wallis test.

**Table 1 ijerph-16-03754-t001:** Geographical and population characteristics of the study regions.

	Region A (Community A, 3653 inhabitants)	Region B1 (Communities < 7000 inhabitants)	Region B2 (Communities ≥ 7000 inhabitants)
	Mean ± SD	Mean ± SD	Mean ± SD
Area of communities (km^2^) ^1^	61.7 ± 0	38.6 ± 22.4	43.1 ± 22.6
Population count ^1^	3653 ± 0	4330 ± 1741	14022 ± 6275
Population density ^1^	59.2 ± 0	145.3 ± 93.9	373.5 ± 165.6
GP density (per 1000 inhabitants) ^2^	0.82 ± 0	0.57 ± 0.41	0.63 ± 0.23
GP supply rate ^2,3^ (%)	114.8	89.2	100.3
Specialist density (per 1000 inhabitants) ^2^	0.27 ± 0	0.25 ± 0.43	1.12 ± 0.99
Number of undersupplied specialist groups ^4,^*	1 out of 13	6 out of 13	
Psychotherapist density (per 1000 inhabitants) ^2^	0.22	0.19	
Psychotherapist supply rate (%) ^2,3^	129.2^4^	115.8	
Dentist density (per 1000 inhabitants) ^5^	0.55	0.90^4^	

Abbreviations: GP, general practitioner. ^1^ Source: Federal Statistical Office of Germany; ^2^ Source: Association of Statutory Health Insurance Physicians; ^3^ Physician supply rate is a measure of the official care planning, based on a corrected relation of physicians/citizens. The planning unit for specialists equals the whole county, whereas the number of GPs is planned on a smaller scale (using geographical units that are smaller than counties but bigger than communities). * The supply rate is calculated for 10 general specialists and three specialist physicians. Reported is the number of specialists available at <110% (required density) [35]; ^4^ Of the entire county including the community; ^5^ Source: Association of Statutory Health Insurance Dentists [40]—special care planning for dentists does not exist in Germany.

**Table 2 ijerph-16-03754-t002:** Characteristics of the study population by region/community.

	Community A (located in Region A, 3653 inhabitants)		Region B1 (Communities with < 7000 inhabitants)		Region B2 (Communities with ≥ 7000 inhabitants)		*p*
Sociodemographic characteristics							
Sex							
Female	921	418 (45.4)	95	39 (41.1)	114	47 (41.2)	0.538 ^b^
Age	921		95		114		0.035 ^a,^*
18–39 years		206 (22.4)		28 (29.5)		34 (29.8)	
40–65 years		462 (50.2)		50 (53.6)		53 (46.5)	
≥66 years		253 (27.5)		17 (17.9)		27 (23.7)	
Monthly income	812		86		94		0.101 ^a^
<1500€		199 (25)		12 (14.0)		19 (20.2)	
1500–<3000€		337 (41.5)		39 (45.3)		39 (41.5)	
≥3000€		276 (34.0)		35 (40.7)		36 (38.3)	
School education/training ^2^	851		92		114		0.525 ^a^
Secondary school (6–9 years)		304 (35.7)		37 (40.2)		38 (35.2)	
Vocational training		272 (32)		30 (32.6)		33 (30.6)	
Master craftsman or university degree		275 (32.3)		25 (27.2)		37 (34.3)	
Currently employed	879	562 (63.9)	93	65 (69.9)	108	64 (59.3)	0.293 ^b^
Health-related characteristics							
Time resources to take care of one’s health ^1^, mean ± SD)	895	2.65 ± 0.87	94	2.56 ± 0.68	114	2.46 ± 0.86	0.230 ^a^
Financial resources to take care of one’s health ^1^ (mean ± SD)	890	1.72 ± 0.72	92	1.55 ± 0.70	112	1.55 ± 0.61	0.013 ^a,^*
Support by friends, neighbors and family, if needed, is etc. ^1^ (mean ± SD)	851	1.479 ± 0.68	91	1.45 ± 0.62	108	1.45 ± 0.72	0.799 ^a^
Self-assessed physical health ^1^	902	1.478 ± 0.61	94	1.44 ± 0.63	112	1.40 ± 0.61	0.261 ^a^
Self-assessed mental health ^1^ (mean ± SD)	887	1.345 ± 0.59	94	1.26 ± 0.55	112	1.3 ± 0.57	0.248 ^a^
Presence of any chronic disease ^2^	892	368 (41.3)	94	33 (35.1)	112	40 (35.7)	0.306 ^b^

Abbreviations: SD, standard deviation. ^a^ Kruskal–Wallis test was used to test for differences between the regions. ^b^ Chi-square test was used to test for differences between the regions; ^1^ Rating: 1 = very good, 2 = good, 3 = satisfactory, 4 = poor, 5 = very poor; ^2^ Defined as “a persistent disease that needs continuous control or treatment” in the questionnaires. * *p* < 0.05.

**Table 3 ijerph-16-03754-t003:** Multinomial logistic regression model for outcome (1) “HCS provided by GPs”.

Characteristics ^1^	Adjusted OR (95% CI)		
	Very Good/Good vs. Very Poor/Poor ^#^	Satisfactory vs. Very Poor/Poor ^#^	Very Good/Good vs. Satisfactory ^#^
*Attribution to region/community*			
Region B2 (≥7000 inh)	0.01 (0–0.03) **	0.21 (0.09–0.49) **	0.06 (0.03–0.12) **
Region B1 (<7000 inh)	0.01 (0–0.03) **	0.18 (0.08–0.42) **	0.07 (0.03–0.14) **
Community A (3653 inh)	Reference		
*Age*			
18–39 years	0.51 (0.19–1.34)	1.52 (0.58–3.95)	0.38 (0.18–0.78) **
40–65 years	0.73 (0.3–1.77)	1.52 (0.58–3.95)	0.48 (0.25–0.92) *
≥66 years	Reference		
*Monthly income (€)*			
0–<1500	3.45 (1.25–9.49) *	3.68 (1.3–10.39) *	0.93 (0.49–1.78)
1500–<3000	1.79 (0.87–3.65)	1.87 (0.88–3.96)	0.95 (0.56–1.61)
≥3000	Reference		
*Evaluation of own financial resources for preventive healthcare*			
Very good/good	2.3 (0.77–6.89)	0.77 (0.25–2.36)	2.96 (1.47–5.94)
Satisfactory	1.09 (0.4–3.02)	0.81 (0.29–2.24)	1.35 (0.74–2.48)
Very poor/poor	Reference		
*Evaluation of support from friends and family*			
Very good/good	2.36 (0.81–6.85)	1.58 (0.53–4.68)	1.49 (0.71–3.09)
Satisfactory	0.82 (0.28–2.38)	1.04 (0.35–3.06)	0.79 (0.37–1.65)
Very poor/poor	Reference		

Abbreviations: HCS, healthcare service; CI, confidence interval; Inh, inhabitants; OR, odds ratio. #Reference; Only variables that were statistically significant in univariate multinomial logistic regression are reported. Excluded variables are sex, education, employment status, self-reported chronic disease, evaluation of traffic infrastructure, evaluation of one’s physical and mental health status. ** *p* < 0.01; * *p* < 0.05.

**Table 4 ijerph-16-03754-t004:** Results of multinomial logistic regression for outcome (2) “HCS provided by specialists”.

Characteristics ^1^	Adjusted OR (95% CI)		
	Very Good/Good vs. Very Poor/Poor ^#^	Satisfactory vs. Very Poor/Poor ^#^	Very Good/Good vs. Satisfactory ^#^
*Attribution to Region/community*			
Region B2 (≥7000 inh)	1.98 (0.68–5.74)	1.07 (0.46–2.47)	1.85 (0.55–6.18)
Region B1 (<7000 inh)	2.54 (1.06–6.09) *	3.28 (1.87–5.75) **	0.77 (0.32–1.85)
Community A (3653 inh)	Reference		
*Age*			
18–39 years	0.64 (0.29–1.37)	0.98 (0.56–1.71)	0.65 (0.28–1.5)
40–65 years	0.52 (0.27–1.02)	0.93 (0.58–1.51)	0.56 (0.27–1.17)
≥66 years	Reference		
*Monthly Income (€)*			
0–<1500	8.22 (2.65–25.48) **	1.34 (0.77–2.32)	6.11 (1.87–19.88) **
1500–<3000	5.41 (1.83–15.94) **	1.08 (0.68–1.69)	5.00 (1.63–15.36) **
≥3000	Reference		

Abbreviations: CI, confidence interval; Inh., inhabitants; OR, odds ratio. #Reference; Only variables that proved to be statistically significant in univariate multinomial regression are reported. Excluded variables are gender, education, employment status, self-reported chronic disease, evaluation of traffic infrastructure, evaluation of one’s physical and mental health status, evaluation of own financial resources for preventive healthcare, evaluation of support from friends and family. ** *p* < 0.01; * *p* < 0.05.

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
