# Peer review of "Patient Input in Regional Healthcare Planning—A Meaningful Contribution"

_ijerph, 2019, doi:10.3390/ijerph16193754_

Round 1

Reviewer 1 Report

Thank you for the privilege of reviewing this timely and germane study.  The authors' writing proficiency, precision, and clarity create an informative and enjoyable learning experience.   

As noted in this study, research suggests patient dissatisfaction with care may be correlated with poor outcomes, mortality, and increased costs (Clarke et al., 2017; Mirzad, Cramm, & Nieboer, 2019; Prentice & Pizer, 2007).  Contemporary literature validates the relationship between lower education, higher age, and higher perceptions of patient satisfaction (Deutsch et al., 2019).

Since the authors acknowledge “relying on quantitative measures alone seems not to be enough to capture regional needs” (p. 10) and “expectations shape satisfaction” (p. 9), it might be appropriate to expand the “future prospects” section to recommend a qualitative approach capable of yielding in-depth detailed data on user perceptions (Rapport et al., 2019) to complement this study's quantitative calibrations.

References:

Clarke, J. L., Bourn, S., Skoufalos, A., Beck, E. H., & Castillo, D. J. (2017). An innovative approach to health care delivery for patients with chronic conditions. Population Health Management20(1), 23-30.

Deutsch, A., Heinemann, A. W., Cook, K. F., Foster, L., Miskovic, A., Goldsmith, A., & Cella, D. (2019). Inpatient rehabilitation quality of care from the patient's perspective: Effect of data collection timing and patient characteristics. Archives of Physical Medicine and Rehabilitation100(6), 1032-1041.

Mirzad, F., Cramm, J. M., & Nieboer, A. P. (2019). Cross-sectional research conducted in the Netherlands to identify relationships among the actual level of patient-centred care, the care gap (ideal vs actual care delivery) and satisfaction with care. BMJ Open9(1), e025147.

Prentice, J. C., & Pizer, S. D. (2007). Delayed access to health care and mortality. Health Services Research42(2), 644-662.

Rapport, F., Hibbert, P., Baysari, M., Long, J. C., Seah, R., Zheng, W. Y., ... & Braithwaite, J. (2019). What do patients really want? An in-depth examination of patient experience in four Australian hospitals. BMC Health Services Research19(1), 38.

Author Response

We thank the reviewers for the highly valuable comments and suggestions. We address individual comments in the attached file and hope that our efforts have improved the document. 

Reviewer 2 Report

This paper needs improvement in many aspects. The findings could be interesting but they are not presented well. Please find my specific comments below.

In the Abstract, the difference between the meaning of a community and a county is unclear. Readers may wonder why the population size of a community is larger than a county. The meaning of the scores is also unclear. The authors should indicate that they are satisfaction scores, and a lower score means better satisfaction (as stated in the methods: 1=very good, 2=good, 3=satisfactory, 4=poor, 5=very poor).

Different recruitment approaches were used between the two regions. The authors should give the rationales. Besides, a bit elaboration about the advantages of the “think aloud” method in pilot testing is needed.

It seems this study aimed to achieve two goals. First, to test for the differences in perception of health services by the citizens in three areas (A, B1 and B2) with different population and healthcare professional densities. Second, to show that the study method can successfully involve patient views in regional healthcare planning. As an original research article instead of a methodology paper, the second aim should be seen as a bonus. The conclusion should highlight the impact of healthcare professional density (relative to population density) on patients’ satisfaction of the services.

Good presentation of the statistics data is important to deliver the key points to the readers and health policy makers. While the tables and charts are quite well presented, the main text of the Results is poorly structured. The results should be organized by themes (with sub-headings), like the way in the Discussion. Relatively, the points in the Discussion are clearer and much better structured. Differences in ratings of different healthcare providers, the association of better satisfaction with older age and lower income are interesting findings. More importantly, the best predictor found was the healthcare provider density. These results were not described clearly in the Results section.

There are many short paragraphs throughout the paper, especially in the Methods section, along with grammatical mistakes.

Author Response

We thank the reviewers for the highly valuable comments and suggestions. We address individual comments in the attached document and hope that our efforts have improved the document.
